# Dissecting the Therapeutic Mechanisms of Sphingosine-1-Phosphate Receptor Agonism during Ischaemia and Reperfusion

**DOI:** 10.3390/ijms241311192

**Published:** 2023-07-07

**Authors:** Georgina C. Wilkins, Jenny Gilmour, Eirini Giannoudaki, John A. Kirby, Neil S. Sheerin, Simi Ali

**Affiliations:** Immunity and Inflammation, Translational and Clinical Research Institute, Faculty of Medical Sciences, Newcastle University, Newcastle upon Tyne NE2 4HH, UK; georgie.wilkins@newcastle.ac.uk (G.C.W.); jenny.gilmour@newcastle.ac.uk (J.G.); giannoue@tcd.ie (E.G.); john.kirby@newcastle.ac.uk (J.A.K.)

**Keywords:** sphingosine 1-phosphate, S1P receptors, endothelial barrier, cell migration, ischaemia–reperfusion injury, chemokine

## Abstract

Sphingosine 1-phosphate (S1P) and S1P receptors (S1PR) regulate many cellular processes, including lymphocyte migration and endothelial barrier function. As neutrophils are major mediators of inflammation, their transendothelial migration may be the target of therapeutic approaches to inflammatory conditions such as ischaemia–reperfusion injury (IRI). The aim of this project was to assess whether these therapeutic effects are mediated by S1P acting on neutrophils directly or indirectly through the endothelial cells. First, our murine model of peritoneum cell recruitment demonstrated the ability of S1P to reduce CXCL8-mediated neutrophil recruitment. Mechanistic in vitro studies revealed that S1P signals in neutrophils mainly through the S1PR1 and 4 receptors and induces phosphorylation of ERK1/2; however, this had no effect on neutrophil transmigration and adhesion. S1P treatment of endothelial cells significantly reduced TNF-α-induced neutrophil adhesion under flow (*p* < 0.01) and transendothelial migration towards CXCL8 during in vitro chemotaxis assays (*p* < 0.05). S1PR1 agonist CYM5442 treatment of endothelial cells also reduced neutrophil transmigration (*p* < 0.01) and endothelial permeability (*p* < 0.005), as shown using in vitro permeability assays. S1PR3 agonist had no effects on chemotaxis or permeability. In an in vivo mouse model of renal IRI, S1PR agonism with CYM5442 reduced endothelial permeability as shown by reduced Evan’s Blue dye extravasation. Western blot was used to assess phosphorylation at different sites on vascular endothelial (VE)–cadherin and showed that CYM5442 reduced VEGF-mediated phosphorylation. Taken together, the results of this study suggest that reductions in neutrophil infiltration during IRI in response to S1P are mediated primarily by S1PR1 signalling on endothelial cells, possibly by altering phosphorylation of VE–cadherin. The results also demonstrate the therapeutic potential of S1PR1 agonist during IRI.

## 1. Introduction

Sphingosine-1-phosphate (S1P) is a biologically active lysosphingolipid generated by the sphingosine kinase (SK)-catalysed phosphorylation of sphingosine. S1P mediates cell survival, proliferation, migration and differentiation, and accomplishes critical tasks in maintaining epithelial and endothelial barrier integrity and governing the migration of immune cells. The pleiotropic physiological effects of S1P are mediated by binding to five cell-surface G-protein-coupled receptors (GPCRs) and sphingosine-1-phosphate receptors (S1PR) 1–5 [1]. S1P receptors couple with heterotrimeric g proteins upon S1P binding, and three different Ga subunits are involved: G_αi_, G_α12/13_ and G_αq_. S1PR1 couples exclusively to G_αi_ to activate small GTPase Rac and cause cell spreading and barrier enhancement downstream. Conversely, S1PR2 and 3 couple with all three, with G_α12/13_ causing RhoA-mediated cellular contraction. The balance between the two pathways determines vascular integrity [2].

The distribution of the receptors on different cell types and the coupling of receptors to different G alpha subunits allow S1P to differentially exert its influence in numerous pathways [3]. S1PR1 is among the most abundant endothelial cell (EC) GPCRs. Loss of S1PR1 signalling in ECs destabilizes adherens junctions, reduces endothelial nitric oxide synthase activity, and increases the expression of leukocyte adhesion molecules [4]. As a consequence, the pharmacological targeting of the S1P/S1P receptor axis has been proposed for the management of several pathophysiological processes, including cancer, chronic inflammation [5], autoimmune disorders, ischaemic stroke [6] and ischaemia–reperfusion injury (IRI) [7].

IRI is a complex and multifactorial complication associated with various pathologies in which blood flow to tissues is disrupted and re-established. A hallmark of IRI is oedema and inflammatory cell infiltration, with neutrophils acting as major mediators of the associated inflammation. Several studies identified S1P among the cell-derived substances locally released during IRI. However, the specific contribution of S1P to the development of IRI is controversial. S1P, as well as the S1P receptor mimetics, including FTY720 and selective S1PR1 agonists, were demonstrated to attenuate IRI in several organs, including the heart [8], brain [9], lungs [10], liver [11] and kidneys [12]. In other circumstances, S1P has been implicated in the propagation of IRI-induced organ damage. For instance, S1PR3 activation was found to potentiate I/R injury in the liver [13], whilst S1PR3 deletion was found to be protective in the kidneys [14]. In addition, the elimination of SK1, which is the major source of local S1P release, reduced the extent of I/R injury in the liver and other organs [5]. Results are conflicting with regard to the role of S1P in IRI and its therapeutic potential.

There are indications that S1P can affect neutrophil migration into tissue and other functions of neutrophils, such as respiratory burst and degranulation [15,16]. S1P also has roles in endothelial cell survival, migration, adherens junction assembly, morphogenesis and barrier integrity. Vascular permeability is a tightly regulated function, and an increase in permeability is an essential component of inflammation. Several in vitro and in vivo studies conducted to date indicate the regulation of endothelial cell permeability by S1P, which has been widely researched in terms of its effect on endothelial cell barrier functions, especially in the context of inflammation. It has been established that following its release, extracellular S1P can signal through its receptors on endothelial cells and cause cytoskeletal rearrangement, specifically actin filament assembly, in order to influence permeability [17,18].

Clearly, S1P signalling is multifaceted and complex in nature, and results regarding the role of S1P in the pathophysiology of IRI are conflicting. This is most likely dependent on cell type and level and the type of receptor expression. Whilst various studies have already shown that increased S1P signalling through S1PR1 can reduce neutrophil infiltration into the site of inflammation during IRI [10,19], we aimed to investigate the relative importance of the role of neutrophils and endothelial cells in mediating this effect. First, a mouse model of peritoneal cell recruitment was used to examine in vivo effects of S1P. Then, the direct effect of S1P on neutrophil adhesion and migration was explored. Furthermore, S1P effects on endothelial cells that may indirectly affect neutrophil migration were investigated. Finally, the role of an S1PR1 agonist in endothelial permeability in an in vivo model of renal IRI was examined.

## 2. Results

### 2.1. Intraperitoneal Treatment with S1P Reduces Neutrophil Recruitment In Vivo

Chemokine CXCL8 is the most potent human neutrophil-attracting chemokine and plays crucial roles in the response to tissue injury. CXCL8 activity inherently depends on interaction with the human CXC chemokine receptors CXCR1 and CXCR2, the atypical chemokine receptor ACKR1 and glycosaminoglycans [20,21]. To determine the effects that S1P pretreatment can have on CXCL8-mediated neutrophil recruitment in vivo, a mouse model of cell recruitment in the peritoneum was used. Mice received either vehicle or S1P as a pretreatment and were then injected with vehicle or CXCL8. Compared with the control groups, which received no CXCL8, neutrophil recruitment was significantly higher in groups that received CXCL8 (Figure 1). The group that received S1P 24 h prior to the CXCL8 injection had decreased neutrophil recruitment in comparison to the group that received vehicle and then CXCL8. In the absence of CXCL8, S1P pretreatment had no effect compared to the mice pretreated with vehicle. Therefore, S1P specifically inhibited CXCL8-induced neutrophil recruitment into the peritoneum.

### 2.2. S1P Signals through S1P Receptors on Neutrophils and Primes Them for CXCL8 Signalling

In vivo results showed that S1P treatment caused a reduction in CXCL8-mediated neutrophil recruitment; therefore, the effect of S1P on neutrophils was interrogated in vitro. First, real-time PCR was used to investigate S1P receptor expression on neutrophils isolated from whole human blood. The S1P receptors S1PR1, S1PR3, S1PR4 and S1PR5 mRNA expression were assessed in parallel with chemokine receptor CXCR1. S1PR1 and S1PR4 were expressed at a higher level than the other receptors relative to CXCR1 expression, with S1PR4 exhibiting the highest level of expression. S1PR3 and 5 were expressed at low levels (Figure 2A). S1PR2 expression was not tested because at the time exon spanning, TaqMan primer/probes were not available.

To investigate the phosphorylation of signalling kinases, Western blotting was used. When neutrophils were treated with CXCL8, there was a significant increase in ERK phosphorylation compared with untreated cells. The extent of ERK phosphorylation was highest after 3 min of CXCL8 treatment, reducing to between 5 and 10 min. Additionally, S1P caused ERK phosphorylation following 10 min of treatment, confirming the capacity of S1P to signal through neutrophils (Appendix A). S1P-induced ERK phosphorylation was lost at 1 h; however, S1P pretreatment prior to CXCL8 stimulation resulted in increased ERK phosphorylation (Figure 2B). This suggests that signalling was not combined but that S1P pretreatment had a priming effect on neutrophils prior to CXCL8 signalling. 

### 2.3. S1P Treatment of Neutrophils Does Not Significantly Alter ICAM-1 and VCAM-1 Adhesion or Transfilter Chemotaxis towards CXCL8

After establishing that S1P can signal through S1P receptors on neutrophils, the effects of this signalling were investigated, namely whether it affected the adhesive and chemotactic abilities of neutrophils. 

The CellixTM system of flow-based adhesion was used to assess whether S1P has an effect on neutrophil adhesion to adhesion molecules, vascular cell adhesion molecule (VCAM)-1 and intercellular adhesion molecule (ICAM)-1. As shown in Figure 3A, when neutrophils were treated with S1P prior to CXCL8 stimulation, no significant effect on VCAM-1 or ICAM-1 adhesion was observed in comparison to neutrophils not treated with S1P, with concentrations of S1P ranging from 0.5–10 μM (data not shown). Next, transfilter chemotaxis assays were utilised in order to assess the effects of S1P treatment on the migratory potential of neutrophils. S1P pretreatment of neutrophils significantly increased neutrophil migration in the absence of a chemotactic agent but not towards CXCL8 (Figure 3B). This indicates that S1P has the capacity to increase the general migratory capabilities of neutrophils independently of CXCL8.

### 2.4. Expression of S1P Receptors on Endothelial Cells

The expression of S1P receptors on HMEC-1 and HUVEC was assessed prior to functional assays. In human umbilical vein endothelial cells (HUVECs), the receptor with the highest relative expression was S1PR1, whilst in HMEC-1. S1PR2 was most highly expressed. HUVEC expressed S1PR1 and 4 to a higher extent than human microvascular endothelial cells (HMEC-1), whilst HMEC-1 expressed S1PR2 and 5 at higher levels. S1PR3 had the lowest expression in both cell types (Figure 4). Using flow cytometry, we also observed the expression of S1PR1 and 3 at the protein level in HMEC-1 cells (Appendix A).

### 2.5. S1P and S1PR 1 and 3 Agonists Are Associated with an Increase in Chemokine Expression in Endothelial Cells

The effect of S1P and associated receptor agonists on chemokine secretion from HUVECs was examined next. Initial experiments were carried out to examine the expression of CXCL8 following stimulation with 1, 5 and 10 μM S1P. Significant expression of CXCL8 was observed at 5 and 10 μM S1P concentrations (data not shown). Hence, 10 μM S1P was used in a chemokine array to determine whether other chemokines were influenced by S1P treatment of HUVECs. Most assessed chemokines were secreted at higher levels by cells treated with S1P compared to untreated cells, with CXCL8 and CCL2 secreted to the highest extent (Appendix A). 

HUVECs were then treated with S1P, S1PR1 agonist CYM5442 or S1PR3 agonist CYM5541 for 4, 8 or 24 h, and CXCL8 secretion was assessed. S1P and CYM5442 caused a significant increase in CXCL8 secretion compared to the control at 4, 8 and 24 h, with S1P causing the greatest increase. CYM5541 and FTY720P had no significant effect (Figure 5). 

### 2.6. S1P Treatment of Endothelial Cells Decreases TNF-α-Induced VCAM-1 Expression, Adhesion and Neutrophil Transmigration through S1PR1 Signalling

To investigate whether S1P could affect neutrophil migration by controlling endothelial cell adhesion molecule expression, HUVECs were treated with S1P, TNF-α or a combination of TNF-α and S1P, and the expressions of ICAM-1 and VCAM-1 were measured by flow cytometry.

Constitutive expression of ICAM-1 was greater than that of VCAM-1. S1P treatment increased the expression of both adhesion molecules. TNF-α caused a greater increase in the expression of both ICAM-1 and VCAM-1; however, when cells were treated with TNF-α and S1P together, VCAM-1 expression was reduced compared to cells treated with TNF-α alone (Figure 6A,B). 

To explore whether the effect of S1P on adhesion molecule expression translates to an effect on neutrophil adhesion to endothelial cells, flow-based adhesion assays were performed. Neutrophils did not adhere to untreated HUVECs. Treatment with S1P caused a small but non-significant increase in adhesion. When HUVECs were treated with TNF-α, there was a significant increase in the adhesion of neutrophils, whereas when treated with a combination of TNF-α and S1P, there was a decrease in adhesion compared with treatment with TNF-α alone (Figure 6C). 

To evaluate whether S1P treatment of endothelial cells can affect neutrophil migration, neutrophil transendothelial chemotaxis experiments were performed. Cells were grown on chemotaxis filters, then treated overnight with S1P, CYM5442, CYM5541 or FTY720P. S1P dose-dependently decreased neutrophil chemotaxis across HUVECs, with significant results at doses of 5 and 10 μM (Figure 6D), and HMEC-1, which was significant at doses of 1, 5 and 10 μM (Appendix A). Treatment with S1P, CYM5442 and FTY720P caused a significant decrease in transendothelial neutrophil chemotaxis towards CXCL8, whereas CYM5541 did not have a significant effect on migration (Figure 6E and Appendix A). Pretreatment of endothelial cells with CYM5442 decreased neutrophil migration to a similar extent as non-selective S1PR agonist FTY720P, indicating the importance of the S1PR1 receptor in the therapeutic potential of S1P. 

### 2.7. Signalling through S1PR1 on Endothelial Cells Reduces Endothelial Barrier Permeability In Vitro and In Vivo and Reduces VE–Cadherin Phosphorylation

HMEC-1 cells were used in an Evan’s Blue (EB) model of endothelial permeability. Monolayers of HMEC-1 cells were grown on permeable filters and either left untreated or treated with CYM54442, CYM5541 or S1P. Permeability to EB–albumin was then assessed. Treatment with CYM5442 for 1 h caused a significant decrease in permeability compared with the untreated cells (Figure 7A). No significant differences in permeability were observed following treatment with CYM5541 or S1P (Figure 7B,C), reinforcing the importance of the S1PR1 pathway in maintaining the endothelial barrier. 

A mouse model of ischaemia–reperfusion injury was used to determine the in vivo effects of an S1P receptor 1 agonist on endothelial permeability during ischaemic damage. CYM5442 or vehicle was administered in mice via intraperitoneal injection prior to and following left kidney pedicle clamping. Permeability to EB was then assessed. Significant decreases in permeability to EB dye were observed in the ischaemic kidneys of mice treated with CYM5442 compared to mice treated with vehicle, indicating the barrier-enhancing effects of the agonist in the context ischaemic injury (Figure 7D).

The molecular basis of S1PR1 agonist-mediated reduction in endothelial permeability was investigated. VEGF treatment increased phosphorylation of Y658, Y685 and Y731 on VE–cadherin compared to untreated cells. CYM5442 treatment also caused phosphorylation, although less than VEGF at both Y658 and Y685. However, when cells were pre-treated with CYM5442, then treated with vascular endothelial growth factor (VEGF), there was a reduction in phosphorylation of Y658 and Y685 compared to VEGF alone. This effect was not seen at Y731.

## 3. Discussion

In this manuscript, we describe the role of SIP in neutrophil migration and, for the first time, show that this is primarily an effect of SIP on endothelial function. This effect is mediated through S1PR1, a receptor that is a potential target for agonist therapy in neutrophil-mediated disease.

Neutrophils are major mediators of many inflammatory conditions, including IRI. Existing literature has already shown the effects of S1P in mitigating neutrophil trafficking during IRI [22,23]. The main aim of this project was to assess whether these therapeutic effects are mediated through S1P acting on neutrophils directly or indirectly through endothelial cells. In a mouse model of peritoneal recruitment, we found a significant increase in neutrophil recruitment in response to CXCL8 injection. S1P injection 24 h before the administration of CXCL8 significantly inhibited neutrophil recruitment into the peritoneal cavity. This supports the concept that S1P can reduce neutrophil migration during injury [22].

In order to address the mechanism, we initially examined the direct effects of S1P on neutrophils, focusing on their migration. Neutrophils isolated from blood mainly expressed S1P receptors S1PR1 and S1PR4, with very low levels of S1PR5, in agreement with Rahaman et al., who found similar receptor expression on neutrophils from healthy subjects [15]. S1P pretreatment appeared to prime neutrophils ahead of CXCL8 signalling, causing enhanced CXCL8-mediated ERK1/2 phosphorylation. This pathway is involved in several neutrophil functions, including adhesion, degranulation, oxidative burst, the formation of neutrophils traps and CXCL8 production [24]. However, neutrophil adhesion under flow conditions was not affected by S1P. This result is consistent with the findings of Kawa et al. [25] but contrasts with a report by Florey and Haskard, who found that S1P enhanced immune-complex-mediated neutrophil adhesion under flow conditions [26].

We found no effect of pretreatment with S1P on neutrophil chemotaxis towards CXCL8, in agreement with Morel et al. [7]. However, we did show that S1P pretreatment of neutrophils increased CXCL8-independent migratory capabilities. Findings regarding the effects of S1P signalling on neutrophils and how these effects translate in vivo are contradictory. A recent study suggested that not only is S1P strongly promigratory but that neutrophil migration into sites of injury is mediated via S1PR2 on neutrophils both in vitro and in vivo [27]. In direct contrast, S1P administration in mice decreased LPS-induced neutrophil migration into the lungs, and in vitro work suggested that this process was mediated through neutrophil-specific S1PR4 [28]. Our data show that although S1P can signal in neutrophils through the S1PR1 and S1PR4 receptors to cause the activation of downstream pathways and prime neutrophils ahead of CXCL8-mediated signalling, this does not translate into an effect on neutrophil adhesion and CXCL8-mediated migration.

To reflect whether the effect of S1P on neutrophil migration in vivo may be the net effect of S1P on other cells, we next examined the effects of S1P on endothelial cells. The barrier-protective effects of S1P on endothelial cells have been corroborated in various studies in vitro [3,29,30,31] and in vivo in a murine model of IRI [32]. We first compared the S1P receptor mRNA expression on primary cells (HUVECs) and the endothelial cell line (HMEC-1). Both cell types expressed high levels of S1PR1; however, there were differences in the level of expression of S1PR2, 4 and 5. S1P has been shown to increase both endothelial [33] and epithelial [34] CXCL8 release, in concordance with our data. S1PR1 appears to be important for increased CXCL8 production, as the use of CYM5422 stimulated expression of CXCL8, although less so than S1P.

Furthermore, S1P can affect adhesion molecule expression by endothelial cells. It was found that the surface expression of both ICAM-1 and VCAM-1 adhesion molecules was increased by S1P. Several studies have also shown that S1P can induce mRNA and/or protein expression of ICAM-1 or VCAM-1 adhesion molecules on endothelial cells [35,36]. It was also interesting that S1P can inhibit TNF-α-induced surface expression of VCAM-1, as supported by Kimura et al., who suggested a S1PR3-specific mechanism [37]. This means that S1P has both stimulatory and inhibitory effects on adhesion molecule expression, most likely involving different S1P receptors. This translated to effects on neutrophil adhesion to endothelial cells under flow conditions, as S1P was shown to reduce TNF-α-induced adhesion. Similarly, Theilmeier et al. showed that S1P can reduce adhesion under the flow of murine macrophages to the TNF-α-activated endothelium [23]. This supports the concept that more specific targeting of receptors is preferable in therapeutic interventions involving S1P and that disease and baseline states may influence the effects of S1P.

The barrier-enhancing effect of S1P is thought to be mediated by S1PR1 signalling both at baseline and in various inflammatory states [38,39,40], in particular during IRI [10,19,32]. Therefore, we examined whether S1P and associated S1PR agonists could reduce neutrophil transmigration via a reduction in monolayer permeability. First, we used neutrophil chemotaxis assays, followed by permeability assays, to link the two. In support of the aforementioned research, S1P and S1PR1 agonist CYM5442 treatment of endothelial cells was found to inhibit neutrophil migration, whilst S1PR3 agonist CYM5541 had no effect. S1PR3 has been reported to have barrier-disrupting effects, causing increased vascular permeability [18], although we did not observe this phenomenon with the S1PR3 agonist used in our study.

Next, we assessed the effects of S1P, CYM5442 and CYM5541 during in vitro permeability assays. Our data revealed that whilst S1P had no significant effect on EB–albumin extravasation through endothelial monolayers, CYM5442 reduced permeability at the baseline level, in accordance with Burg et al. [38]. Our in vivo experiments showed that agonist treatment caused a significant decrease in permeability to EB dye in the ischaemic kidney and a trend towards a decrease in permeability in the non-ischaemic kidney. This is the first study to explore the effect of agonist CYM5442 in the context of IRI in vivo. As seen in vitro, CYM5442 was shown to enhance endothelial integrity; however, there were no significant differences in permeability between the ischaemic kidneys and non-ischaemic kidneys in either treatment group. Our data suggest that endothelial cells are the predominant cell type that mediates the therapeutic effects of S1P during IRI through endothelial S1PR1 signalling. Studies utilising endothelial-specific S1PR1 knockout mice help to support our hypotheses. Perry et al. showed that endothelial S1PR1 was vital in suppressing inflammation during IRI [41], whilst Burg et al. showed increased vascular leak and oedema, as well as increased polymorphonuclear leukocyte numbers (PMNs) in BAL fluid following lung RAR [38]. These results illustrate the importance of endothelial cells specifically, as well as endothelial S1PR1 signalling.

As for a mechanism for the barrier-enhancing effects S1P exhibits, the current study was focused on VE–cadherin, which is an important molecule of adherens junctions, controlling permeability and transcellular cell migration. It has been found in the past that S1P can increase localization of VE–cadherin and other adherens junction molecules such as β-catenin at cell–cell contact regions, assisting in the assembly of adherens junctions without affecting total VE–cadherin protein levels [42,43]. Although several studies have investigated VE–cadherin abundance at cellular junctions, very few have examined the effect of S1P/S1PR agonists on VE–cadherin phosphorylation. VE–cadherin phosphorylation causes subsequent internalization and ubiquitination, ultimately reducing VE–cadherin at intercellular junctions and increasing junction instability. Accordingly, we demonstrated that endothelial cells treated with VEGF showed an increase in phosphorylation of Y658 and Y685, as demonstrated in other studies [44,45,46]. Whilst Scotti et al. demonstrated that S1P treatment reduces VE–cadherin phosphorylation [47], our study went further in identifying S1PR1 signalling as the predominant driver of this phenomenon. When cells were pr-treated with CYM5442, then treated with VEGF, phosphorylation at Y658 and Y685 was reduced compared to VEGF alone. This suggests that barrier enhancement is mediated through dynamic mechanisms involving a reduction in phosphorylation at specific sites on VE–cadherin.

In conclusion, the results of the current study suggest that the therapeutic potential of S1P in the treatment of IRI lies in its direct effect on the endothelium rather than on neutrophils. The reduction in neutrophil recruitment to inflammatory sites is mediated by structural changes to the endothelium, which reduces permeability upon administration of S1P via S1PR1 signalling. These alterations to endothelial permeability occur via reductions in phosphorylation of VE–cadherin. This enhanced barrier results in decreased neutrophil chemotaxis despite increases in the neutrophil chemokine CXCL8 and adhesion molecule expression in endothelial cells (Figure 8). Therefore, S1PR1 may be the most useful target to reduce vascular leakage in pathologies such as IRI.

## 4. Materials and Methods

### 4.1. Animals

Female BALB/c mice (Charles River, Edinburgh, UK) aged 7–10 weeks were used for intraperitoneal recruitment experiments. Male C57BL/6 mice (Charles River, UK) (22–27 g) aged 10–12 weeks were used for IRI experiments due to the greater susceptibility of male mice to IRI [48]. All animal procedures were performed in accordance with UK Home Office project licence PF7050EF5 for intraperitoneal experiments and PD86B3678 for the ischaemia–reperfusion injury model.

### 4.2. Intraperitoneal Recruitment Experiment

Mice were injected intraperitoneally (ip) with either 500 μM S1P or vehicle. We used Huzzah S1P (Avanti Polar Lipids, USA), a human serum albumin (HSA)/S1P complex. Huzzah S1P consists of S1P conjugated with HSA, a physiologically relevant carrier protein, at a 2:1 ratio. Huzzah S1P was reconstituted in sterile PBS with an appropriate S1P concentration (usually 500 μM). Huzzah control (Avanti Polar Lipids, Alabama, USA) was utilised as vehicle for the experiments that used Huzzah S1P, and it was essentially HSA of the same grade as in Huzzah S1P. Then, 24 h later, mice were injected with 5 μg CXCL8 or vehicle. This concentration and time were based on our previous study with FTY720-P [49]. After 4 h, mice were culled by cervical dislocation under anaesthesia, and the abdominal cavity was lavaged three times with 1ml PBS and 3mM EDTA. Lavage samples were centrifuged at 500g for 5 min and resuspended in 125 µL buffer. Peritoneal cells were stained in the same tube with appropriate directly conjugated primary antibodies. Single stains were used as compensation controls for multicolour flow cytometry, as well as “Fluorescence Minus One” (FMO) controls, which contained all antibodies expect one, to place negative gates. The following antibodies were used: anti-mouse CD3ε, CD19, CD11b, CD11c, Ly6G and F4/80 (Biolegend, London, UK). The Zombie Aqua Fixable Viability Kit viability dye (Biolegend, UK) was used to distinguish alive cells from dead cells. Alive cells were gated on the CD3/CD19 axis to exclude T and B lymphocytes. The CD3/CD19 population was further gated on the Ly6G/CD11b axis, where neutrophils were gated as a Ly6Ghi CD11bhi population. Flow cytometry was performed on a FACSCanto II instrument (BD Biosciences, NJ, USA), and data were analysed using FlowJo 7.6 software (Treestar).

### 4.3. Ischaemia–Reperfusion Injury Model

During the induction of IRI, the core temperature of the animal was maintained at 37 °C by enclosing the mouse in a warmed pad with continuous measurement of rectal temperature. Using this method, we observed significant injury at the shorter of the tested time points. Our group previously published data on ischaemia–reperfusion injury (IRI) (the left renal pedicle was clamped for 25, 35 or 45 min, and the kidneys were allowed to reperfuse for 24 h). We observed a significant increase in neutrophils following 25 min of ischaemia and 24 h of reperfusion [50].

Mice were treated with 3 mg/kg CYM5442 or an equivalent volume of dimethyl sulfoxide (DMSO) as a vehicle control via ip injection 1 h prior to surgery. The left renal artery and vein were clamped for 30 min. Then, 23 h following the beginning of surgery, mice were again treated with 3 mg/kg CYM544 ip. One hour later, EB dye (20 mg/kg) was injected into the lateral tail vein. After 30 min, mice were sacrificed, and the kidneys were removed. One-quarter of each kidney was weighed and placed 1:5 weight (mg) to volume (µL) in 50% trichloroacetic acid (TCA) in 0.9% saline and homogenised using a Tissue Lyser II. The samples were then centrifuged and quantified by fluorescence (ex620nm em680nm).

### 4.4. Cell Culture

HMEC-1 cells (ATCC^®^ CRL-3243TM, USA) were cultured in complete MCDB 131 medium (Thermofisher Scientific, Horsham, UK) containing 10% foetal bovine serum (FBS, Lonza, Basel; Switzerland), L-glutamine dissolved in sterile water supplemented with 15.7 mL/L sodium bicarbonate solution (7.5% *w*/*v*), 10 ng/mL epidermal growth factor (EGF); 1 μg/mL hydrocortisone, 100 μg/mL penicillin and 0.1 mg/mL streptomycin (all Sigma Aldrich, Dorset, UK). HUVECs were cultured in complete endothelium growth medium (ATCC, Virginia, USA) containing Endothelial Cell Growth Kit-BBE (ATCC, USA). Human colorectal adenocarcinoma cells (Caco-2, ATCC^®^ HTB-37TM, USA) were cultured in high-glucose Dulbecco’s Modified Eagle’s Medium (DMEM, Sigma Aldrich, USA) containing 10% FBS, L-glutamine dissolved in sterile water with 15.7 mL/L sodium bicarbonate solution (7.5% *w*/*v*), 100 μg/mL penicillin and 0.1 mg/mL streptomycin (all Sigma Aldrich, USA). Human primary neutrophils were isolated using Percoll density gradient and suspended in RPMI-1640 medium with 0.5% fatty-acid-free BSA, 2 mM L-glutamine, 100 U/mL penicillin and 0.1 mg/mL streptomycin (all Sigma Aldrich, USA).

### 4.5. RT-qPCR

RNA was isolated with the TRI reagent (Sigma-Aldrich) method. Reverse transcription for cDNA synthesis was performed using Agilent’s AffinityScript Multiple Temperature cDNA synthesis kit. The cDNA was then used for detection of gene expression by real-time PCR using TaqMan^®^ Gene Expression assays for detection of genes (18S, Hs99999901_s1; S1PR1, Hs00173499_m1; S1PR3, Hs01019574_m1; S1PR4, Hs02330084_s1; S1PR5, Hs00928195_s1; CXCR1, Hs01921207_s1; Applied Biosystems, Woolston, UK), along with TaqMan^®^ Gene Expression Master Mix 2X (Applied Biosystems) in an Applied Biosystems qPCR machine, with 18S used as the reference gene and data presented relative to CXCR1 mRNA expression.

### 4.6. Immunoprecipitation

HMEC-1 cells were treated with 100 ng/mL VEGF for 30 min, 10 µM CYM5442 for 1 h or CYM5442 for 1 h followed by VEGF for 30 min or left untreated. Following treatment, cells were pelleted and lysed as for Western blot. Lysate was incubated with anti-VE–cadherin (or control) Dynabeads (Thermofisher, Horsham, UK) overnight. Immunoprecipitated proteins were visualised by Western blotting.

### 4.7. Western Blotting

For neutrophil experiments, cells were treated with or without 1µM S1P for 1 h before lysis with lysis buffer (CelLytic M; Sigma-Aldrich, USA) containing protease inhibitors (cOmplete Protease Inhibitor Cocktail; Roche, Switzerland) and phosphatase inhibitors (Halt Phosphatase Inhibitor Cocktail; Pierce, Thermo Scientific, Massachusetts, USA). Protein was measured using a bicinchoninic acid (BCA) protein assay kit (Pierce, Thermo Scientific, USA).

Lysates or immunoprecipitated proteins were resolved using an SDS-PAGE gel before transfer to a polyvinylidene fluoride (PVDF) membrane. The membrane was blocked and probed with primary antibody overnight at 4 °C using either a phospho-ERK1/2 antibody (Thr202/Tyr204; Cell Signalling, Massachusetts, USA), phosphor-VE–cadherin (Y658; Thermofisher, Horsham, UK), phosphor-VE–cadherin (685, Abcam, Boston, USA) or phosphor-VE–cadherin (Y731; Thermofisher, Horsham, UK). After washing the membrane, anti-rabbit HRP-conjugated antibody (Thermofisher, Horsham, UK) was added and incubated for 1 h. Chemiluminescence was visualised using either a Licor Odyssey Fc or X-ray film. Membranes were then stripped and probed again with an appropriate loading control antibody (total ERK1 (Cell signalling, Massachusetts, USA) or total VE–cadherin (Abcam, USA). Densitometry was quantified using ImageJ.

### 4.8. Chemotaxis

For transfilter chemotaxis assays, isolated human neutrophils pretreated with or without 1 μM S1P for 30 min were added to the top chamber of 3 μm transwells. For transendothelial chemotaxis assays, isolated neutrophils were added to transwells containing HUVEC monolayers treated with 0.1–10 μM S1P, 10 μM CYM5442, 10 μM CYM5541 or 10 μM FTY720P overnight or left untreated. Neutrophils were left to migrate towards different concentrations of CXCL8 (R&D Systems) for 90–120 min as indicated. For all experiments, cells that passed through to the bottom chamber were measured by flow cytometry using counting beads (CountBright™ Absolute Counting Beads, Invitrogen, Paisley, UK ).

### 4.9. Flow-Based Adhesion Assay

Vena8 biochips were coated with 10 μg/mL of either recombinant ICAM-1 or VCAM-1 Fc chimera protein overnight (R&D Systems), then blocked with 1% bovine serum albumin (BSA) for 30 min. Primary human neutrophils were resuspended in serum-free RPMI-1640 medium at a density of 1 × 106 cells/mL and treated with 5 μM S1P or left untreated for 1 h. The Cellix system was prepared as previously described [51]. Then, 100 μL of cell suspension (100,000 cells) was added to the biochip mounted on the microscope stage. Prior to being added to the channel, cells were stimulated with 25 ng/mL CXCL8 for VCAM-1-coated channels or 100 ng/mL for ICAM-1-coated channels. Flow was initiated at a shear stress of 10 dyne/cm2 for 10 s to diffuse the cells into the channel, then continued at 0.5 dyne/cm2 for 5 min to assess adhesion. Images of 4–8 different fields of view were captured along the length of each channel. Adherent cells were counted in each captured image.

To assess adhesion to endothelial cells, VenaEC chips were coated with a HUVEC monolayer and treated overnight with 10 μM S1P, 100 ng/mL TNF-α or both. Then, 2 × 10^5^ carboxy fluorescein succinimidyl ester (CFSE)-stained primary human neutrophils in serum-free medium (2 × 10^6^ cells/mL) were added to each channel, and flow was initiated as described above. Images were captured in both bright-field and fluorescent mode for each field of view (4–9 fields along the length of the channel). Adherent neutrophils were counted in fluorescent images, using the bright-field images for verification.

### 4.10. Measurement of Chemokine Concentration

The Human Chemokine Array Kit (R&D Systems, Abingdon, UK) was used to measure chemokine concentration in supernatants of HUVECs treated for 24 h with 10 μM S1P or left untreated. CXCL8 release by HUVECs treated with 10 µM S1P, 10 μM CYM5442, 10 μM CYM5541 or 10 μM FTY720P at sequential timepoints or left untreated was measured by an ELISA assay (PeproTech, Cheshire, UK).

### 4.11. Flow Cytometry

Endothelial cells were treated overnight with 10 μM S1P, 100 ng/mL TNF-α or 100 ng/mL TNF-α + 10 μM S1P or left untreated. Following treatment, adherent cells were detached and suspended in FACS buffer (2% FBS in PBS) with 1 mM EDTA. Fc receptor blocking was performed, and cells were incubated with directly conjugated primary antibodies ICAM-1 and VCAM-1 (BioLegend) for 45 min at 4 °C. After washing and resuspension in FACS buffer, acquisition was performed in an FACS Canto II (BD Biosciences, California, USA). Analysis was carried out using FlowJo 7.6.

### 4.12. Permeability Assays

Monolayers of HMEC-1 or Caco-2 were grown on 0.4 μm transwells. HMEC-1 cells were then treated with 10 µM S1P, 10 µM CYM5442 or 10 µM CYM5541 for 1 h or left untreated. Then 20 mM of Evans Blue (EB) conjugated to albumin was added to the upper chamber of each transwell. After 30 min, 50 μL samples were removed from the lower chamber of each well and added to a 96-well plate before reading with a plate reader at an absorbance of 640 nm.

### 4.13. Statistical Analysis

All statistical analysis was performed using GraphPad Prism 8. Error bars included on graphs are the standard error of the mean unless otherwise specified. The statistical tests are described for each experiment individually. Statistical significance is indicated as follows: * < 0.05; ** < 0.01; *** < 0.005; **** < 0.001.

## Figures and Tables

**Figure 1 ijms-24-11192-f001:**
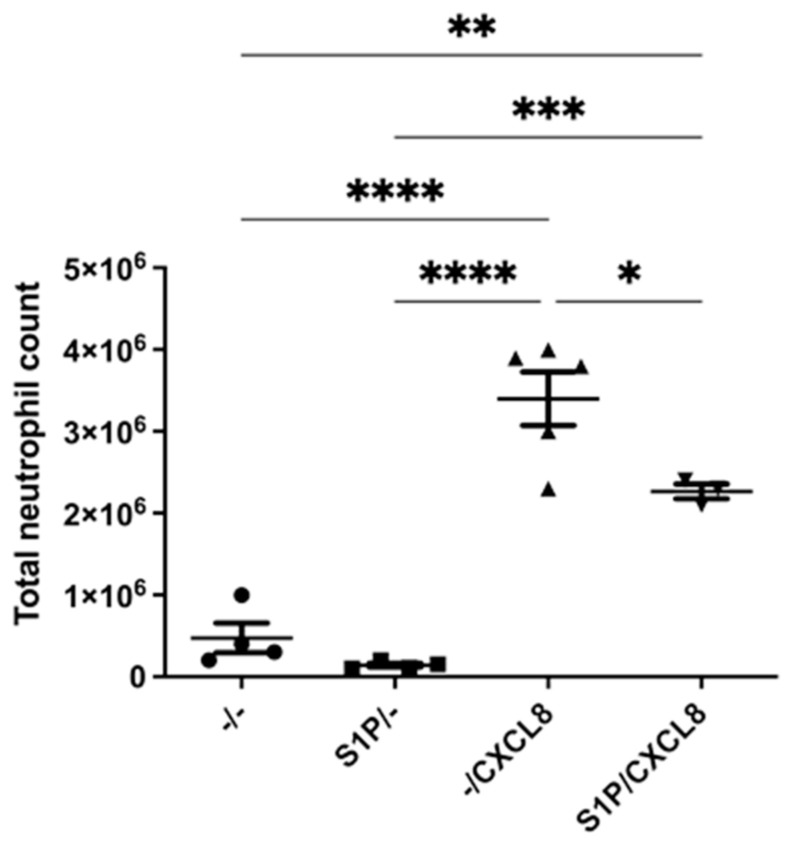
S1P affects CXCL8-mediated cell recruitment. The effects of S1P on CXCL8 recruitment in the peritoneum of female BALB/6 mice. Mice were administered 200 μL or 500 μM S1P or vehicle via intraperitoneal injection. After 24 h, they were injected with 200 μL 5μg CXCL8 or vehicle. After 4 h, the mice were euthanised, and the peritoneal fluid was collected. Cells were stained for use in flow cytometry and counted using counting beads. Total cell count and neutrophil count (Ly6Ghi CD11bhi) were measured. *n* = 4 (PBS), *n* = 4 (S1P/-), *n* = 5 (-/CXCL8), *n* = 4 (S1P/CXCL8). Statistical analysis was performed using a one-way ANOVA with Tukey’s post test. * = *p* < 0.05, ** = *p* < 0.01, *** = *p* ≤ 0.001, **** = *p* < 0.0001.

**Figure 2 ijms-24-11192-f002:**
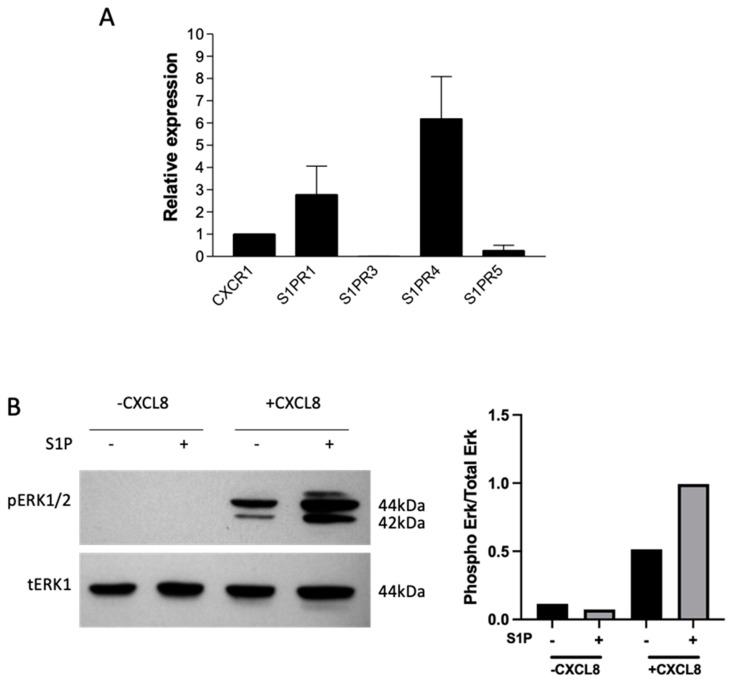
S1P signals through neutrophils and primes them for CXCL8 signalling. Neutrophils were isolated from whole human blood and used immediately in experiments. (**A**) Relative mRNA expression of CXCR, S1PR1, S1PR3, S1PR4 and S1PR5 in neutrophils. mRNA expression was detected using real-time PCR, and data were normalised to housekeeping gene 18S and presented relative to CXCR1 mRNA expression; *n* = 2. (**B**) Effect of S1P pretreatment on ERK1/2 phosphorylation in neutrophils induced by CXCL8. Neutrophils were pretreated with (+) or without (-) 1µM S1P in serum-free media for 1 h, then treated with (+) and without (-) 100 ng/mL CXCL8 for 3 min. Neutrophils were lysed and used in Western blotting experiments with pERK1/2 and total ERK as a loading controls. Densitometry was quantified, and the band density of pERK is presented as a ratio of total ERK. Data are representative of two independent experiments.

**Figure 3 ijms-24-11192-f003:**
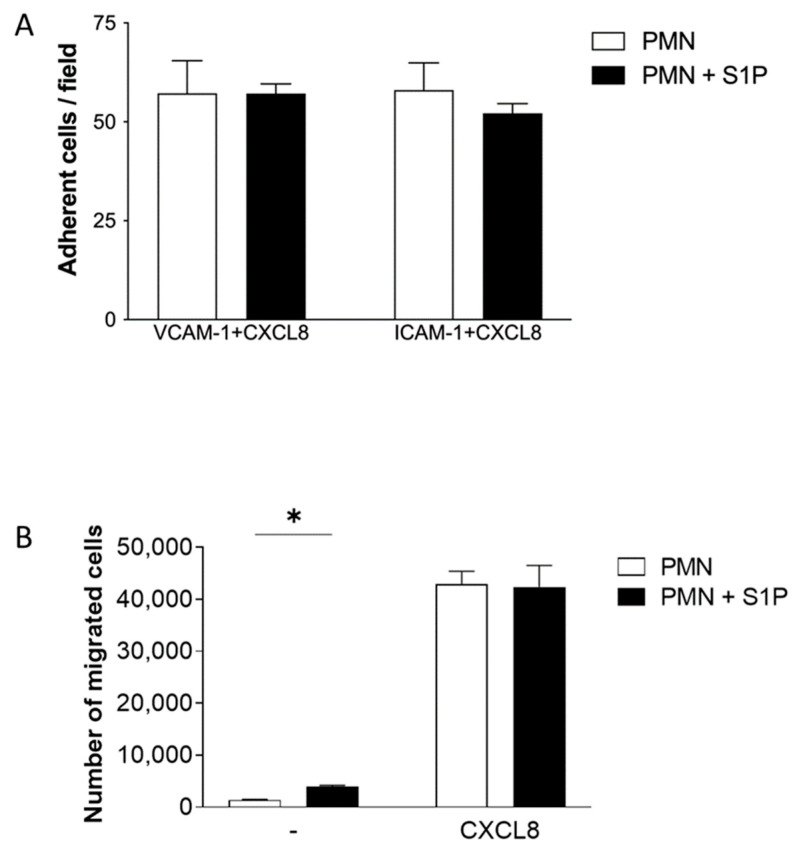
S1P pretreatment of neutrophils had no effect on adhesion and migration towards CXCL8. (**A**) Effect of S1P pretreatment on CXCL8-stimulated flow-based adhesion. Neutrophils were pretreated with or without 5µM S1P in serum-free media for 1 h, then stimulated with 25 ng/mL CXCL8 for VCAM-1 or 100 ng/mL for ICAM-1. Cells were then used in VCAM-1 or ICAM-1 flow-based adhesion assays; *n* = 3. (**B**) Effect of S1P pretreatment on neutrophil chemotaxis. Neutrophils were incubated for 30 min with or without 1 μΜ S1P in serum-free media, then allowed to migrate towards vehicle (-) or 50 ng/mL CXCL8. Total migrated cells were counted by flow cytometry; *n* = 3. Statistical analysis was performed using a two-tailed unpaired student’s *t*-test; * = *p* < 0.05.

**Figure 4 ijms-24-11192-f004:**
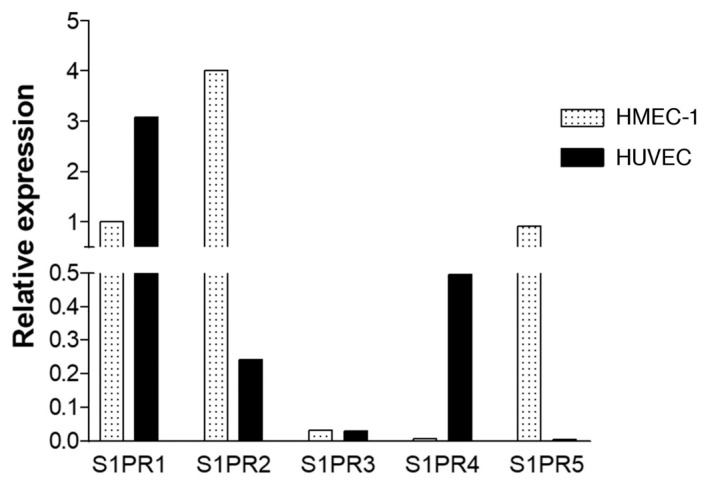
Cell line HMEC-1 and primary HUVEC show distinct S1P receptor expression. Relative mRNA expression of S1PR1, S1PR2, S1PR3, S1PR4 and S1PR5 in HMEC-1 and HUVECs. mRNA expression was detected using real-time PCR and data normalised to housekeeping gene GAPDH and presented as expression relative to that of S1PR1 by HMEC-1. Representative of three independent experiments.

**Figure 5 ijms-24-11192-f005:**
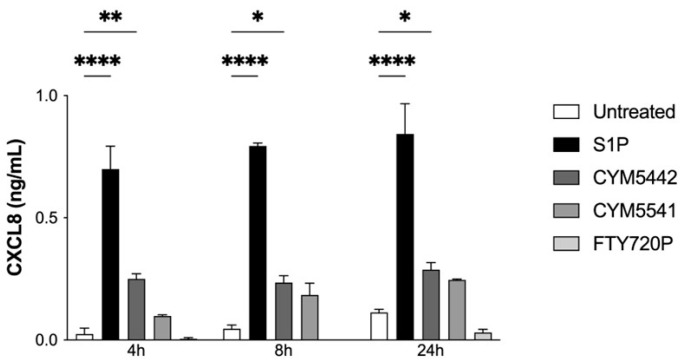
S1P receptor agonists induce chemokine production in endothelial cells. CXCL8 production by HUVECs. HUVECs were treated with 10 µM S1P, 10 μM CYM5442, 10 μM CYM5541 or 10 μM FTY720P or left untreated for 4, 8 or 24 h. Supernatants were collected for use in a CXCL8 ELISA assay; *n* = 2. Statistical analysis was performed using a two-way ANOVA with Dunnet’s post test. * = *p* < 0.05; ** = *p* < 0.01; **** = *p* < 0.0001.

**Figure 6 ijms-24-11192-f006:**
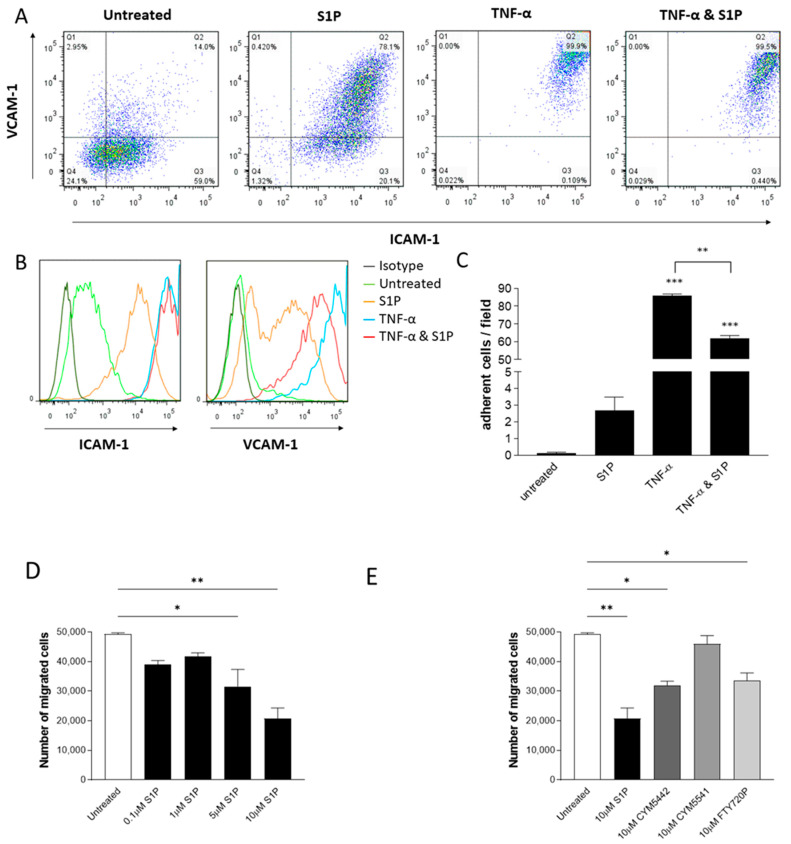
S1P treatment of endothelial cells reduces neutrophil adhesion and transmigration. (**A**,**B**) ICAM-1 and VCAM-1 surface expression in HUVECs treated with S1P and TNF-α. HUVEC were treated overnight with 10 μM S1P, 100 ng/mL TNF-α or 100 ng/mL TNF-α + 10 μM S1P or left untreated. The cells were then stained with directly conjugated monoclonal antibodies for ICAM-1 and VCAM-1 or appropriate isotype controls and analysed by flow cytometry. (**A**) Dot plots showing the effects of S1P, TNF-α or TNF-α + 10 μM S1P treatment on ICAM-1 and VCAM-1 expression in HUVECs. Data are representative of three independent experiments. (**B**) Histograms showing the effects of S1P, TNF-α or TNF-α + 10 μM S1P treatment on ICAM-1 and VCAM-1 expression in HUVECs. Dark green: isotype control; light green: untreated; orange: S1P; blue: TNF-α; red: TNF-α + S1P. Data are representative of three independent experiments. (**C**) Neutrophils were isolated from whole human blood, stained with CFSE and used in a flow-based assay of adhesion to HUVECs that had been treated with 10 μM S1P or 100 ng/mL TNF-α; *n* = 3. (**D**) Neutrophil transendothelial migration towards CXCL8. HUVECs were seeded on chemotaxis filters and treated overnight with 0.1–10 µM S1P. Neutrophils isolated from whole human blood were added to the top chamber of the chemotaxis filters and left to migrate for 2 h towards 10 ng/mL CXCL8. Total migrated cells were counted using counting beads; *n* = 2 (**E**) Neutrophil transendothelial migration through HUVECs treated with S1P analogues. HUVECs were seeded on chemotaxis filters and treated overnight with 10 µM S1P, CYM5442, CYM5541 or FTY720P. Neutrophils isolated from whole human blood were added to the top chamber of the chemotaxis filters and left to migrate for 2 h towards 10 ng/mL CXCL8. Total migrated cells were counted using counting beads; *n* = 2. Statistical analysis was performed using a one-way ANOVA with Tukey’s post test. * = *p* < 0.05; ** = *p* < 0.01; *** = *p* < 0.001.

**Figure 7 ijms-24-11192-f007:**
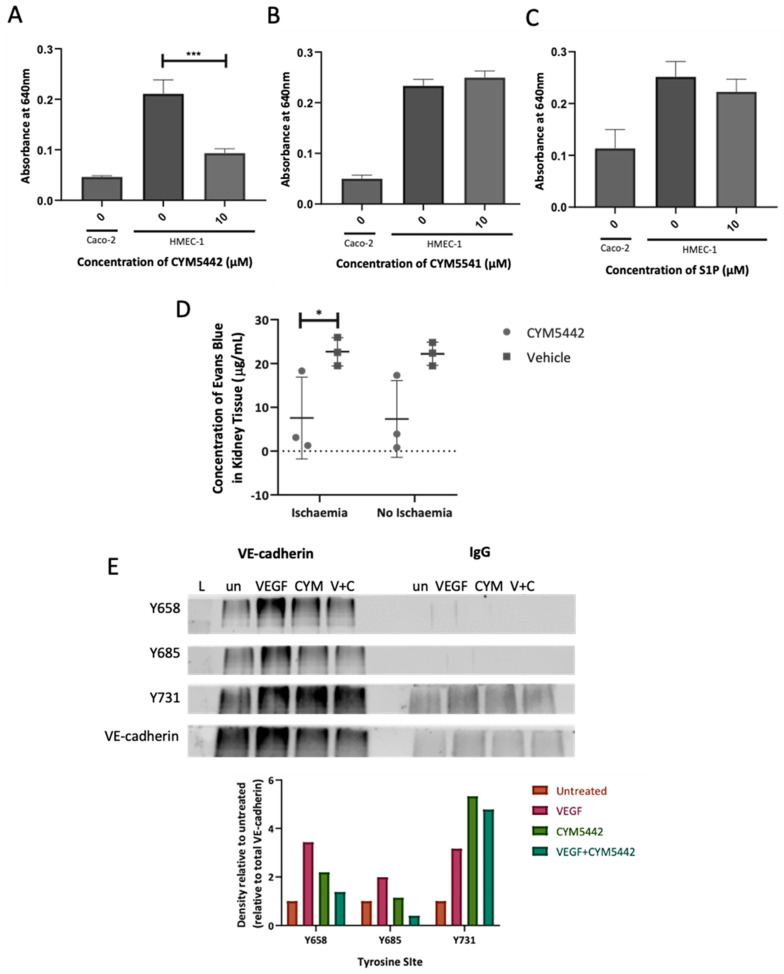
An S1P receptor 1 agonist reduces endothelial barrier permeability. (**A**–**C**) HMEC−1 cells were seeded on chemotaxis filters and left to grow for 7 days. Cells were then treated with (**A**) 10 µM CYM5442, (**B**) 10 µM CYM5541 or (**C**) 10 µM S1P for 1 h or left untreated; then, 20 mM EB was added to the top transwell chamber of. After 30 min, samples were removed from the lower chamber and read at 640 nm. Caco-2 cells were used as negative controls for all experiments; *n* = 3. (**D**) Male C57BL/6 mice were injected with 3 mg/kg CYM5442 or vehicle one hour prior to surgery. The kidneys of these mice were clamped for 30 min, and the mice were left to recover for 23 h. The mice were then injected with 3 mg/kg CYM5442, then with 20 mg/kg EB dye 1 h later. Thirty min later, the mice were sacrificed. One-quarter of each of the harvested kidneys was homogenised, and the supernatant was read at ex620 nm em680 nm; *n* = 3. Statistical significance was determined by a two-way ANOVA. (**E**) VEGF-mediated phosphorylation of VE–cadherin on Y658, Y685 and Y731. HMEC-1 cells were serum-starved overnight, then left untreated or treated with 100 ng/mL VEGF for 30 min, 10 µM CYM5442 for 1 h or 10 µM CYM5442 for 1 h followed by 100 ng/mL VEGF for 30 min. The cells were lysed and incubated with antibody-coated dynabeads overnight (VE–cadherin or IgG control). Western blots for immunoprecipitated proteins were probed with antiphospho-VE–cadherin Y658, Y685 or Y731 antibodies. The lower panel shows total immunoprecipitated VE–cadherin after stripping and probing the same blot. Labelling on blots: L, protein ladder; un, untreated; VEGF, treated with 100 ng/mL VEGF for 30 min; CYM, 10 μM CYM5442 for 1 h; V + C, CYM5442 for 1 h followed by 100 ng/mL VEGF for 30 min. Densitometry was performed and the band density of phospho-VE–cadherin is presented as a ratio of total VE–cadherin before presenting values relative to untreated cells. Data are representative of three independent experiments. * = *p* < 0.05; *** = *p* < 0.001.

**Figure 8 ijms-24-11192-f008:**
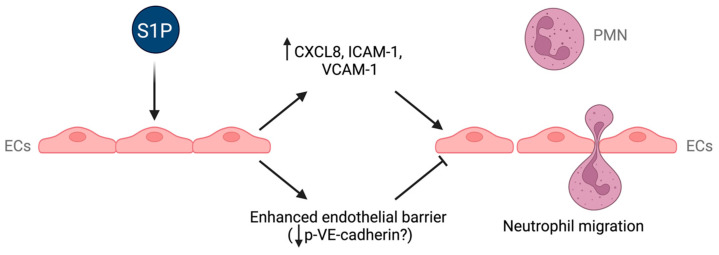
Effects of S1P on endothelial cells. Schematic diagram summarizing the diverse effects of S1P on endothelial cells and how these effects could affect neutrophil migration. ECs: endothelial cells; p-VE–cadherin: phosphorylated VE–cadherin; PMN: neutrophil.

## Data Availability

All data are available in the main text or the Appendix A.

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
