# Peer review of "Dissecting the Therapeutic Mechanisms of Sphingosine-1-Phosphate Receptor Agonism during Ischaemia and Reperfusion"

_ijms, 2023, doi:10.3390/ijms241311192_

Round 1

Reviewer 1 Report

Dissecting the therapeutic mechanisms of sphingosine-1-phosphate receptor agonism during ischaemia-reperfusion injury

Knowing that sphingosine 1-phosphate (S1P) and S1P receptors (S1PR) regulate many cellular processes 10 including lymphocyte migration and endothelial barrier function it is very important to asses if their transendothelial migration could be the target of therapeutic approaches  to inflammatory conditions such as ischaemia reperfusion injury (IRI).

In this manuscript we describe the role of SIP on neutrophil migration and for the 289 first time show that this is primarily an effect of SIP on endothelial function. This effect is mediated through S1PR1 and identifies this receptor as a potential target for agonist therapy in neutrophil mediated disease. 

Methods used and the experimental model in mice are well described as well as the figures. References must be up-dated. 

Author Response

My co-authors and I were greatly encouraged by the editor’s invitation to modify our manuscript to address the generally constructive points raised by two reviewers. The specific changes we have made in response to comments from each of the reviewers are detailed below:

Reviewer 1:

We would like to thank the reviewer for their positive comments. We have now updated the reference list.

Reviewer 2 Report

The authors of the present study investigated to understand the S1P receptor agonism in the IRI mouse model. In this study, the authors used both in vitro and in vivo systems to understand S1P and its receptor, particularly S1PR1, role in neutrophils transmigration and endothelial permeability via VE-cadherin. I believe the study has serious experimental flaws with very low "n" numbers in both in vitro and in vivo experiments and difficult-to-follow results. Not sure about the rationale for using different gender for different in vivo experiments. There is no quantification data for western blots and not sure why the authors used only T-ERK1 to normalize p-ERK1/2. Furthermore, the observed mechanism by the S1P or S1PR1 agonist is not fully convincing. 

Extensive English language editing required. 

Author Response

My co-authors and I were greatly encouraged by the editor’s invitation to modify our manuscript to address the generally constructive points raised by two reviewers. The specific changes we have made in response to comments from each of the reviewers are detailed below:

Reviewer 2:

  1. Not sure about the rationale for using different gender for different in vivo experiments.

Male mice were used for IRI experiments due to their greater susceptibility to IRI renal injury (1).

For peritoneal recruitment model there is no obvious gender bias. Furthermore, there is better availability of female BALB/c mice (Charles River, UK) aged 7-10 weeks, hence they were used for this set of experiments.

  1. There is no quantification data for western blots and not sure why the authors used only T-ERK1 to normalize p-ERK1/2.

Below we show quantification data for figure 2B (rebuttal figure 1) and 7E (rebuttal figure 2). If needed, we can include these figures in supplementary data. Total ERK is usually considered a reliable indicator for the normalisation of the level of phosphorylation of pERK1/2. On hindsight we could have used another housekeeping protein as well e.g., actin/GAPDH.

Please figures and references in the attached file.

Reviewer 3 Report

The authors tried to investigate a direct effect of S1P on neutrophil adhesion and migration and demonstrated that S1P effects on endothelial cells by inducing chemokine CXCL8, reducing endothelial barrier permeability  that indirectly affect neutrophil migration.

Some concerns:

1.    CXCL8 needs to be introduced in the “Introduction” or in Result 2.1, to explain the specific reason to choose CXCL8 as neutrophil chemotaxis, why not CXCL1, CXCL2, or CCL2?  Simply indicate “CXCL8 mediated neutrophil recruitment...” is not sufficient.

2.    This reviewer has different interpretation of the result 2.2 . At first, in result 2.1 the authors claim that S1P specifically inhibited CXCL8 induced neutrophil recruitment into the peritoneum and speculate that S1P pre-treatment had a priming effect on neutrophils prior to CXCL8 signaling. 3 min p-ERK peak after CXCL8 treatment on neutrophils is high likely nonspecific activation which terminate within 5 min. in figure 2B, a kinetic stimulation (0, 5, 10, 20 min) with CXCL8 or S1P is necessary. It would be interesting to know if there is alteration of PI3K-Akt signaling.  S1PRs expression has to be measured in protein level, such as WB or flow cytometry. CXCL8 has receptors CXCR1, CXCR2, and only CXCR1 has been measured in mRNA level. This reviewer would suggest investigating whether S1P stimulation on neutrophils alters CXCR1, or CXCR2 expression levels, and the decreased levels CXCR1/2 might be the cause of the inhibited CXCL8 induced neutrophil recruitment.

3.    Figure 2 and 3: what does “-“mean?  Figure7E looks like mislabeled on top.

Author Response

My co-authors and I were greatly encouraged by the editor’s invitation to modify our manuscript to address the generally constructive points raised by the reviewers. The specific changes we have made in response to comments from each of the reviewers are detailed in the file attached.

Reviewer 4 Report

The article is devoted to the actual topic - to develop of new pharmacological targets to treat the inflammatory component of ischemic-reperfusion kidney injury. The article presents data on the anti-inflammatory effect of sphingosine-1-phosphate receptor agonism in interleukin-induced inflammation, as well as the effect of this substance on kidney permeability at the ischemia-reperfusion injury. The overall study design has logic; on the whole, the methods are adequate to the set research tasks. However, some points do not allow to understanding the results objectively.

1) Title of this article is not relevant to content. In the title, it should be noted that the researchers studied the inflammatory response at all and injury of the kidneys to ischemia-reperfusion. This is the only fundamental remark.

Below I will give minor comments, the corrections of which I leave to the discretion of the authors.

2) Introduction: Should note the type of GPCR (see doi: 10.1091/mbc.E17-03-0136)

3) Methods and design: Author used 4 model of inflammation and model of kidney permeability as a method of ischemia-reperfusion damage detection. Uses of agonists and antagonists are adequate. However, if the authors intended to leave “kidney injury” in the title, they should have provided more classic evidence of kidney damage – serum levels of markers of kidney damage, histological pictures. Otherwise, the word "damage" should be removed from the title of the article.

4) In Figure 7D, row data on the effect of S1PR3 agonist CYM5442 show no difference with and without ischemia on renal permeability to EB dye. The number of observations in this experiment in vivo raises questions. Perhaps due to the small number of observations, ANOVA does not show an adequate result. The data should be rechecked and the effect of S1PR3 agonist on intact kidney permeability explained.

Author Response

My co-authors and I were greatly encouraged by the editor’s invitation to modify our manuscript to address the generally constructive points raised by the reviewers. The specific changes we have made in response to comments from each of the reviewers are detailed below and in the file attached.

Reviewer 4

  1. Title of this article is not relevant to content. In the title, it should be noted that the researchers studied the inflammatory response at all and injury of the kidneys to ischemia-reperfusion. This is the only fundamental remark.

We appreciate the reviewers comments but struggle to understand point 1. As the “only fundamental remark”, we endeavour to make the change suggested if some clarity could be provided.

  1. Introduction: Should note the type of GPCR (see doi: 10.1091/mbc.E17-03-0136)

We agree with the reviewers’ comments and have added a section into the introduction and cited the suggested paper.

  1. Methods and design: Author used 4 model of inflammation and model of kidney permeability as a method of ischemia-reperfusion damage detection. Uses of agonists and antagonists are adequateHowever, if the authors intended to leave “kidney injury” in the title, they should have provided more classic evidence of kidney damage – serum levels of markers of kidney damage, histological pictures. Otherwise, the word "damage" should be removed from the title of the article.

We agree with the reviewers comment and have removed ‘injury’ from the manuscript title

  1. In figure 7D, row data on the effect of S1PR3 agonist CYM5442 show no difference with and without ischemia on renal permeability to EB dye. The number of observations in this experiment in vivo raises questions. Perhaps due to the small number of observations, ANOVA does not show an adequate result. The data should be rechecked and the effect of S1PR3 agonist on intact kidney permeability explained.

We agree that the sample size of our in vivo work is a limitation of this study and that perhaps the lack of significance between s1pr1 agonist treated and untreated for the intact kidney could be due to this. Our group has previously published data on ischemia reperfusion injury (IRI) (left renal pedicle was clamped for 25, 35 or 45minutes and kidneys allowed to reperfused for 24h). We observed a significant increase in neutrophils following 25 minutes of ischemia and 24h of reperfusion (3).

We think that our finding of a significant increase in vascular integrity following cym5442 treatment compared to untreated in our ischaemic kidney group is notable. It will be difficult to carry out any further repeats of the experiment in the time frame of the rebuttal.

Round 2

Reviewer 2 Report

The revised version of this manuscript is not fully answered my suggestion and did not see any major changes in the revised manuscript. 

Reviewer 3 Report

The authors ignored this reviewer's suggestions, not even in letter or "Responses to the reviewers".

Reviewer 4 Report

Manuscript can be accept in present form

Author Response

Please see the attachmnet
